

# Influence of common assumptions regarding aerosol composition and mixing state on predicted CCN concentration

Manasi Mahish[1], Anne Jefferson[2,3], and Don R. Collins[1]

[1]Department of Atmospheric Sciences, Texas A&M University, College Station, Texas, USA.
[2]Cooperative Institute for Research in Environmental Science (CIRES), University of Colorado, Boulder, CO, USA.
[3]NOAA Earth System Research Laboratory, Boulder, CO, USA

*Correspondence to*: Don R. Collins (dcollins@tamu.edu)

**Abstract.** A 4-year record of aerosol size and hygroscopic growth factor distributions measured at the Department of Energy's SGP ARM site in Oklahoma, U.S. were used to estimate supersaturation (S)-dependent cloud condensation nuclei concentrations ($N_{CCN}$). Baseline or reference $N_{CCN}(S)$ spectra were estimated by using the data to create a matrix of size- and hygroscopicity-dependent number concentration (N) and then integrating for S > critical supersaturation ($S_c$) calculated for the same size and hygroscopicity pairs using $\kappa$-Köhler Theory. The accuracy of those estimates was assessed through comparison with the directly measured $N_{CCN}$ at the same site. Subsequently, $N_{CCN}$ was calculated using the same dataset but with an array of simplified treatments in which the aerosol was assumed to be either an internal or an external mixture and the hygroscopicity either assumed or based on averages derived from the growth factor distributions. The CCN spectra calculated using the simplified treatments were compared with those from the baseline approach to evaluate the impact of commonly used approximations. Among the simplified approaches, assuming the aerosol is an internal mixture with size-dependent hygroscopicity parameter ($\kappa$) resulted in estimates closest to those from the baseline approach over the range in S considered.

## 1 Introduction

Aerosol particles activate to form cloud droplets when they encounter a supersaturation, S, that exceeds their size- and composition-dependent critical supersaturation, $S_c$. Those particles with $S_c$ less than the relevant local, instrumental, or prescribed S are called cloud condensation nuclei or CCN. Particles composed of soluble inorganic species are usually more hygroscopic than those composed of organic species and thus are more efficient nuclei. For particles of any composition, $S_c$ decreases and CCN "activity" increases with increasing particle size. Thus, variation in both the size distribution and the chemical composition of an aerosol significantly affects CCN concentration.

Particle $S_c$ can be determined for inorganic species using Köhler Theory, provided the physico-chemical properties (e.g., size, hygroscopicity) of the solutes are known (Köhler, 1936). However, atmospheric aerosols frequently contain a significant amount of organic material as well (Hallquist et al., 2009; Pennington et al., 2013). In several studies, particles composed



entirely of organic species have been reported to be largely ineffective in droplet formation (Abbatt et al., 2005; Prenni et al., 2007). But the solubility and surface tension-reducing properties of the organic component can sometimes have significant influence on $S_c$ (e.g., Bigg, 1986; Roberts et al., 2002; Raymond, 2002, 2003; Chan et al., 2008; Smith et al., 2008; Yli-Juuti et al., 2011). Extended Köhler Theory can reasonably predict CCN concentration with knowledge of the size distribution and

chemical composition of a multi-component aerosol (Raymond, 2002, 2003; Bilde and Svenningsson, 2004; Hartz et al., 2006; Svenningsson et al., 2006). Introduction of a single hygroscopicity parameter by Petters and Kreidenweis (2007) has simplified description and comparison of hygroscopicity and CCN activity for particles composed of single or multiple inorganic and organic species (Moore et al., 2011).

The CCN concentration is also dependent on aerosol mixing state. Previous closure studies have shown that assumption of an internal mixture generally results in an overestimate of the CCN concentration and assumption of an external mixture in an underestimate (Covert et al., 1998; Chuang et al., 2000; Mircea et al., 2002; Rissler et al., 2006; Roberts et al., 2006; Furutani et al., 2008; Kuwata et al., 2008; Shantz et al., 2008; Bougiatioti et al., 2009; Chang et al., 2010; Kammermann et al., 2010; Roberts et al., 2010; Rose et al., 2010; Wang et al., 2010, Moore et al., 2012; Gácita et al., 2017). However, assumption of

either mixing state leads to reasonable results for aged aerosols (Ervens et al., 2010). While inclusion of mixing state (Broekhuizen et al., 2006; Cubison et al., 2008; Lance et al., 2009; Zaveri et al., 2010; Padró et al., 2012) and chemical composition (Medina et al., 2007; Stroud et al., 2007; Cubison et al., 2008; Gunthe et al., 2009; Murphy et al., 2009; Bhattu and Tripathi, 2015) can increase the accuracy with which CCN concentration can be estimated, both can be highly variable with time and with particle size, and are often unavailable with current measurement techniques and not easily incorporated

into aerosol descriptions used in models. Moreover, chemical composition and mixing state are greatly simplified in large scale models, e.g., inorganic/organic, internal/external. Supporting such treatment, several studies have shown that $N_{CCN}$ is most sensitive to the aerosol size distribution (Conant et al., 2004; Dusek et al., 2006), and the assumption of internal mixing has resulted in fairly accurate predictions (Liu et al., 1996; Cantrell et al., 2001; VanReken et al., 2003; Rissler et al., 2004; Chang et al., 2007; Wang et al., 2008; Gunthe et al., 2009). Ervens et al. (2007, 2010) reported that description of the mixing

state is relatively more important than that of the size-resolved chemical composition. Other studies suggest detailed information of size distribution, chemical composition, and mixing state is important for achieving closure among aerosol and CCN measurements (Mircea et al., 2005; Medina et al., 2007; Stroud et al., 2007; Quinn et al., 2008; Lance et al., 2009; Asa-Awuku et al., 2011; Almeida et al., 2014; Che et al., 2016).

In this study, size-resolved concentration and subsaturated hygroscopicity measurements made by the U.S. Department of Energy's (DOE) Atmospheric Radiation Measurement (ARM) program were used to estimate CCN concentration using an array of assumptions for composition and mixing state. Baseline CCN spectra ($N_{CCN}$ vs. S) were first derived by using i) each combined set of size and hygroscopicity distributions and ii) $\kappa$-Köhler Theory to create a pair of matrices describing i) N and ii) $S_c$ as a function of particle dry diameter, $D_d$, and hygroscopic growth factor, GF. The resulting $N_{CCN}$ calculated by integrating



the N matrix over all elements for which S > $S_c$ was compared to direct measurements to evaluate the consistency of the datasets and the accuracy of the estimation technique. The spectra calculated using alternate approaches and assumptions were then compared to those from the baseline approach and the results were used to consider the scatter and bias introduced with simplifications commonly employed in large scale models.

## 2 Site description and measurements

The data were recorded at the Southern Great Plains (SGP) central facility (CF1) (36° 36' 18.0" N, 97° 29' 6.0" W), located in a mixed land use area of cattle pastures and agricultural fields (mainly wheat, hay and corn) near Lamont, Oklahoma, U.S. The climate at the site is continental with hot and humid summers and cool winters. The site is impacted by air masses originating from several regions, with accompanying diversity in aerosol concentration and properties. The chemical composition of the aerosol found at the site is complex and highly variable with time (evident in Figures S1 and S2) and with particle size. The size dependence of aerosol composition as reflected in that of GF was described by Mahish and Collins (2017). Table 1 lists the routine aerosol measurements at the site that were used for the analysis presented here. All datasets used for this analysis are available for download from the ARM archive. Data from the scanning mobility particle sizer (SMPS) / hygroscopicity tandem differential mobility analyzer (HTDMA) system were used for most of the analyses described here. That instrument sequentially measures a size distribution and then a set of hygroscopic growth factor distributions at 7 dry particle sizes every ~45 min. Details of the SMPS/ HTDMA system and processing of the data it generates are available in the Tandem Differential Mobility Analyzer/Aerodynamic Particle Sizer (APS) Handbook (Collins, 2010) and in the work of Gasparini et al. (2004).

**Table 1.** List of instruments, measured quantities, manufacturer, and year installed

| Instrument | Measurement | Manufacturer / Model | Installation year |
|---|---|---|---|
| Scanning Mobility Particle Sizer (SMPS; part of the "TDMA" system) | Size distribution from 0.012 to 0.74 μm dry diameter ($D_d$) | Fabricated, Texas A&M University | 2005 |
| Hygroscopic Tandem Differential Mobility Analyzer (HTDMA; part of the "TDMA" system) | Hygroscopic growth factor distributions of 0.013, 0.025, 0.05, 0.1, 0.2, 0.4, and 0.6 μm $D_d$ particles at 90% RH | Fabricated, Texas A&M University | 2005 |
| Cloud Condensation Nuclei counter (CCNc) | CCN concentration at a fixed set of supersaturations | Droplet Measurement Technologies CCN-100 | 2009 |
| Condensation Particle Counter (CPC) | Concentration of $D_d$ > 0.01 μm particles | TSI Inc. 3010 | 1996 |
| Aerosol Chemical Speciation Monitor (ACSM) | Sub 1-μm chemical composition (organics, sulfate, nitrate, ammonium, and chloride) | Aerodyne Research, Inc. | 2010 |




### 3 Screening and time interval selection

Data from each instrument were validated separately and periods having erroneous data or no data were excluded from analysis. Data from time periods during which instrument problems or failure was evident or when one of the following occurred were not used:

    a)    The total particle concentration ($N_{CN}$) calculated by integrating the SMPS size distribution differed significantly from that directly measured with the CPC,

    b)    The $N_{CCN}$ measured with the CCNc exceeded the $N_{CN}$ measured by the CPC, possibly due to malfunction of the CCNc,

    c)    The sample flow entering the upstream (1st) DMA had an RH>30%, or

    d)    The sample flow entering the downstream (2nd) DMA had an RH< 85%.

The categorized data quality during the period of analysis is shown in Figure 1.

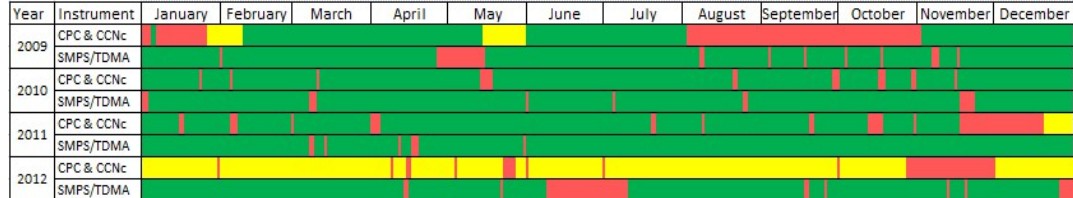

**Figure 1:** Data quality during the analysis period. Periods during which data are available and no significant problems were
identified are green, those during which confidence in at least some subset of the data is low are yellow, and those during which data are unavailable or thought to be erroneous are red.

### 4 Use of $\kappa$-Köhler Theory

For each of the approaches used to estimate $N_{CCN}$, the calculation of $S_c$ from measured particle size and measured or assumed hygroscopicity employed $\kappa$-Köhler Theory (Petters and Kreidenweis, 2007). Aqueous particles as measured in the HTDMA
at 90% RH and those in a supersaturated environment were assumed to be ideal solutions. This assumption was made largely out of necessity because size-resolved composition measurements are not available and the bulk submicron measurements made with the ACSM show the aerosol composition is complex and varies considerably during the year. Specifically, there is a strong seasonality in the soluble inorganic content, with sulfate dominant from roughly April through October and nitrate dominant from November through March, as is evident in the sulfate:nitrate ratio shown in Figure 2. Thus, the choice of soluble
inorganic component(s) needed to model the extent and effect of solution non-ideality would vary by month over the 4-year period of this analysis, as well as over shorter periods of days or even hours accompanying changes in the origin and processing of the sampled aerosol. Furthermore, any attempt to model the aerosol and cloud droplets as non-ideal solutions would require



consideration of the influence of the significant organic content at the site. As shown in Table 2, averaged throughout the year organics contribute over 50% to the total submicron mass concentration.

**Table 2.** Mass concentration fraction from ACSM measurements at SGP

| Year | Mass concentration fraction (%) | | |
| --- | --- | --- | --- |
| | Total organics | Ammonium sulfate | Ammonium nitrate |
| 2011 | 57 | 17 | 26 |
| 2012 | 56 | 18 | 26 |
| 2013 | 56 | 24 | 20 |

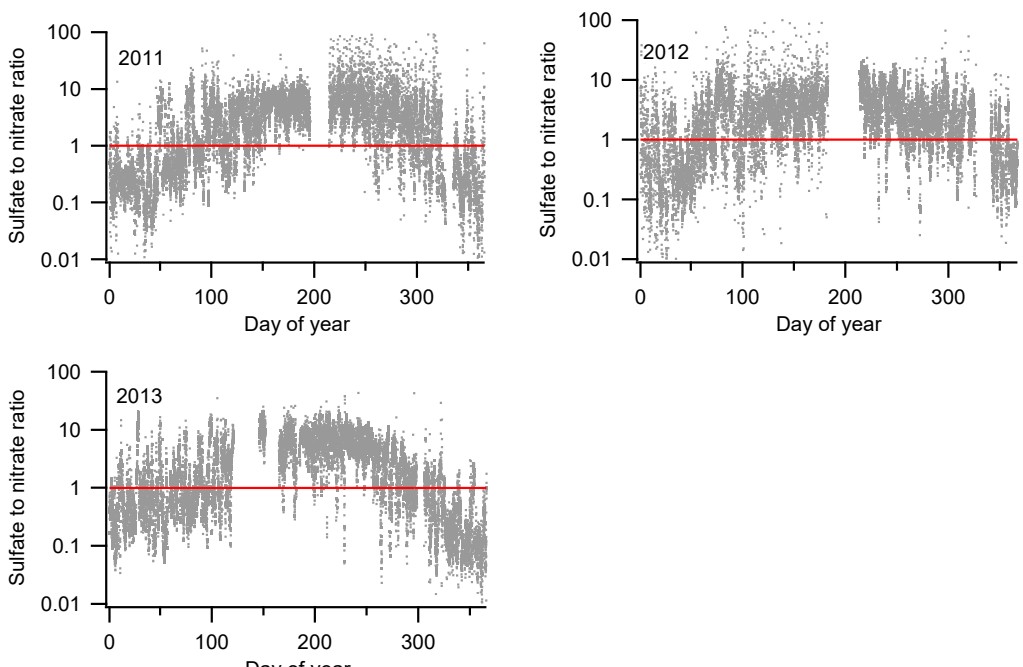

**Figure 2:** Sulfate to nitrate equivalent concentration ratio based on ACSM chemical composition data from 2011(top, left), 2012 (top, right), and 2013 (bottom). Equivalent concentration $= \frac{\text{molar concentration}}{\text{equivalence factor}}$, where the equivalence factor for sulfate

10  and nitrate are 0.5 and 1, respectively.





For all CCN concentration estimate approaches for which the HTDMA data were used, GF was related to $\kappa$ using the following expression from Petters and Kreidenweis (2007)

$$\kappa = ([GF(RH)]^3 - 1)\frac{\left(\exp\left(\frac{A}{GF(RH)\cdot D_d}\right)-RH\right)}{RH} \tag{1}$$

$$A = \frac{4\cdot\sigma_{sol}\cdot M_w}{R\cdot T\cdot\rho_w}$$

Where $M_w$ and $\rho_w$ are the molecular weight and liquid density of water, respectively. The solution surface tension, $\sigma_{sol}$, was assumed to be that of pure water, 0.072 J m⁻². Because aerosol and cloud droplet aqueous solutions are assumed to be ideal it is not necessary to know the contributions of different aerosol components to the overall hygroscopicity for the internal mixture calculations. But for assumed external mixtures the hygroscopicity of two or more particle types must be determined or assumed. For this analysis, particle types assumed to be present in external mixtures were i) particles composed of soluble

inorganics, ii) particles composed of soluble organics, and iii) particles composed of insoluble (and non-hygroscopic) components. The hygroscopicity parameter of the soluble inorganic particles, $\kappa_{inorg}$, was assumed to be 0.6, which is similar to that of ammonium sulfate and ammonium nitrate. As aerosol organic components at the SGP site are not well characterized, direct derivation of $\kappa_{org}$ is not possible. Here, a $\kappa_{org}$ value of 0.1 was estimated using the mixing rule (Petters and Kreidenweis, 2007)

$$\kappa_{org} = \frac{V_{overall}\kappa_{overall} - \left\{V_{overall}-\left(\frac{m_{org_{NR}}}{\rho_{org_{NR}}}+\frac{\frac{b_{abs_{BC}}}{\beta_{BC}}}{\rho_{BC}}\right)\right\}\kappa_{inorg}}{\frac{m_{org_{NR}}}{\rho_{org_{NR}}}} \tag{2}$$

The total particle volume concentration ($V_{overall}$) was calculated from the measured size distribution. The submicron average hygroscopicity parameter ($\kappa_{overall}$) was calculated as the volume concentration-weighted average $\kappa$, calculated from the measured size and hygroscopicity distributions. The mass concentration of non-refractory organics ($m_{org_{NR}}$) was measured by the ACSM and that of black carbon, BC, was calculated as the ratio of measured submicron particle light absorbance ($b_{abs_{BC}}$)

and an assumed absorption efficiency at 0.55 µm wavelength of 7.5 m² g⁻¹ (Yang et al., 2009). The density, $\rho$, of non-refractory organics and BC were both assumed to be 1.3 g cm⁻³ (Nakao et al., 2013).

The resulting seasonal profiles of $\kappa_{overall}$ and $\kappa_{org}$ in 2011 are shown in Figure 3. Unlike the $\kappa_{org}$ profile, $\kappa_{overall}$ was highest in the winter and lowest in the summer. The high wintertime $\kappa_{overall}$ is a result of high concentrations of inorganic compounds,

especially nitrate, while the relatively low summertime $\kappa_{overall}$ is caused by higher organic mass concentrations, as shown in Figure 4. Positive Matrix Factorization (PMF) analysis provided in the Organic Aerosol Component (OACOMP) ARM Value Added Product (VAP) (Fast et al., 2013) indicates that less-hygroscopic biomass burning organic aerosol (BBOA) was prevalent from February through April in 2011, thus lowering $\kappa_{org}$ in winter and spring. Aged SOA (MO-OOA), which is moderately hygroscopic, was more abundant in summer, and thus raised $\kappa_{org}$. The $\kappa_{org}$ in the spring and fall lies between that



of the winter and summer. Although some seasonal variation in $\kappa_{org}$ is evident in Figure 3, the assumed average value of 0.1 is reasonable for this study.

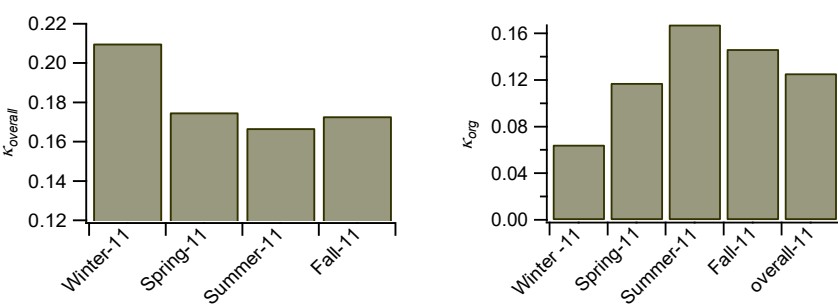

**Figure 3:** Seasonal profile of $\kappa_{overall}$ (left) and $\kappa_{org}$ (right) in 2011.

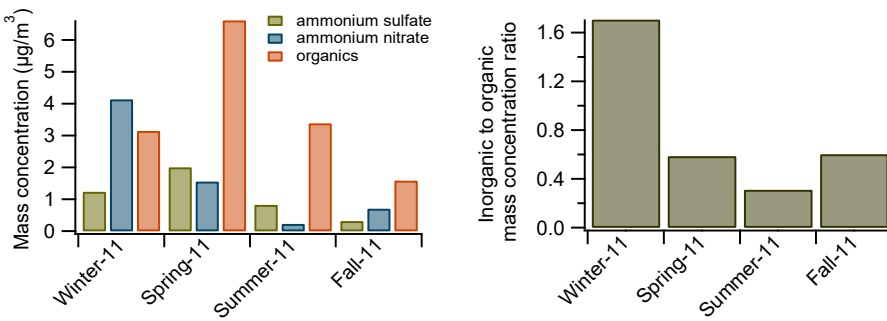

**Figure 4:** Seasonal average of inorganic and organic mass concentration in 2011.

**5 Description of models used for estimating $N_{CCN}$**

To simplify both the comparison of the varied approaches used to estimate $N_{CCN}$ and the comparison of the results, a common

framework will be used to describe all of the approaches even though more straightforward descriptions would suffice for many of them. For all approaches a CCN spectrum, $N_{CCN}(S)$, was calculated for each size distribution measured by the SMPS. For the simplest approaches the HTDMA data were not considered and a fixed hygroscopicity parameter was assumed. For all others the GF distributions were interpolated and extrapolated to each of the 90 size bins in the size distribution measurements and then converted to $\kappa$ distributions using Equation 1. The differences among those approaches arise from the use and (any)

averaging of the $\kappa$ distributions, as summarized in Figure 5.





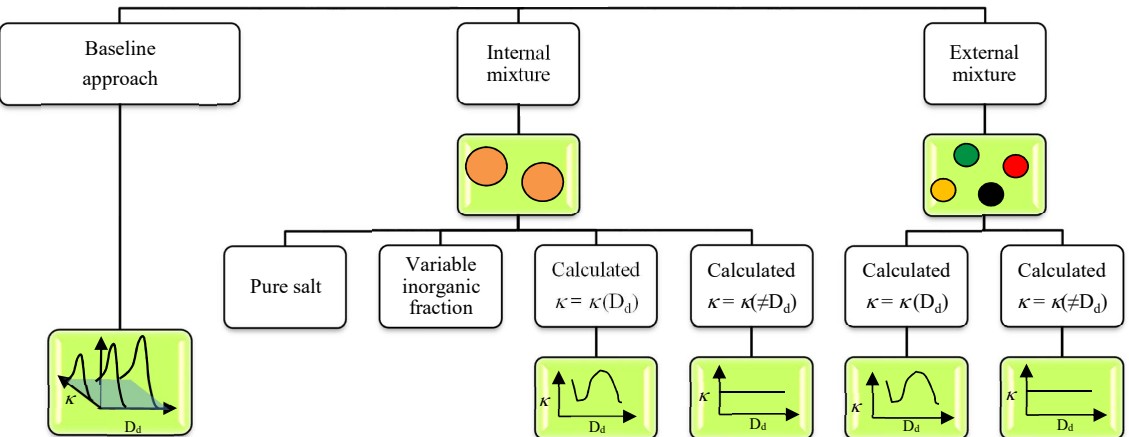

**Figure 5:** Overview of the $N_{CCN}$ calculation approaches.

As a baseline for comparison with estimates from other approaches and with direct measurements, $N_{CCN}$ was calculated using
the full GF distributions without any averaging. Because all of the information in the GF distributions is retained, these
estimates are expected to be more accurate than those from any of the other approaches, all of which rely on averaged or
assumed hygroscopicity. For each measurement sequence the interpolated $\kappa$ distributions were combined with the size
distribution to create a matrix of number concentrations as a function of $D_d$ and $\kappa$, such as that shown graphically in Figure 6.
A Köhler curve relating equilibrium S to droplet diameter, D, was calculated for each ($D_d$, $\kappa$) pair using Equation 3 below. The
$S_c$ for each pair was calculated as the maximum value of equilibrium S along the curve.

$$S(D) = \frac{D^3 - D_d^3}{D^3 - D_d^3(1-\kappa)} \exp\left[\frac{4M_w\sigma_{sol}}{RT\rho_w D}\right] - 1 \tag{3}$$

As with calculations using Equation 1, $\sigma_{sol}$ was assumed to be that of water, 0.072 J m$^{-2}$. The result can be viewed as a matrix
with elements of $S_c$ and the same $D_d$ and $\kappa$ arrays as used in the number concentration matrix described above. $N_{CCN}$ was
estimated for a prescribed S by integrating the number concentration N($D_d$, $\kappa$) for which the S > $S_c$. This is presented
graphically in Figure 6, with $N_{CCN}$ calculated by summing the N elements (whole or part) above and to the right of one of the
four curves, each of which connects the elements having the same $S_c$. The resulting $N_{CCN}$ estimates were first compared with
direct measurements made with the CCNc and then with the results of the other estimate approaches outlined below.





For all other $N_{CCN}$ estimates the aerosol was assumed to be either an internal mixture or an external mixture, as is generally required for regional and global scale climate models. The goals here were to assess the error introduced when making these simplifying assumptions and to identify the approach(es) most suitable for an aerosol similar to that found at SGP.

**5.1 Assumed internal mixtures**

To treat the aerosol as an internal mixture the $D_d$-dependent $\kappa$ distributions described above were replaced with a single $\kappa$ value that is either dependent on $D_d$ ($\kappa=\kappa(D_d)$) or the same for all $D_d$ ($\kappa\neq\kappa(D_d)$). The former comes simply from the number concentration-weighted average of the $\kappa$ distributions at each $D_d$, with the result depicted in Figure 6 in the same manner as for the baseline approach matrix. For approaches for which $\kappa$ is assumed to be size independent, it was calculated either as the

average of $\kappa(D_d)$ or, neglecting the hygroscopicity measurements, as that of particles composed of 20%, 50%, or pure soluble inorganics (~ammonium sulfate, AS) by volume ($\kappa = 0.12, 0.30$, and $0.60$, respectively).

**5.2 Assumed external mixtures**

External mixtures were assumed to have up to three particle types: insoluble, $\kappa = 0.0$, organic, $\kappa_{org} = 0.1$, and inorganic, $\kappa_{inorg} = 0.6$. As with the assumed internal mixture approaches, both size dependent and size independent scenarios were considered.

For both, independent size distributions of the different particle types were calculated from the average $\kappa$ ($=f(D_d)$ or $\neq f(D_d)$) using Equations 4 and 5 below.

Case 1:  for $\kappa(D_d) > \kappa_{org}$

$$\left(\frac{dN}{dlogD_d}\right)_{inorg} = \frac{\kappa(D_d)-\kappa_{org}}{\kappa_{inorg}-\kappa_{org}}\left(\frac{dN}{dlogD_d}\right)_{SMPS}$$

$$\left(\frac{dN}{dlogD_d}\right)_{org} = \left(\frac{dN}{dlogD_d}\right)_{SMPS} - \left(\frac{dN}{dlogD_d}\right)_{inorg} \tag{4}$$

Case 2:  for $\kappa(D_d) \leq \kappa_{org}$

$$\left(\frac{dN}{dlogD_d}\right)_{org} = \frac{\kappa(D_d)}{\kappa_{org}}\left(\frac{dN}{dlogD_d}\right)_{SMPS} \tag{5}$$

$$\left(\frac{dN}{dlogD_d}\right)_{insoluble} = \left(\frac{dN}{dlogD_d}\right)_{SMPS} - \left(\frac{dN}{dlogD_d}\right)_{org}$$

The result is depicted graphically in Figure 6, where the lower and higher horizontal lines represent the organic and inorganic particle types, respectively. As with the other approaches, $N_{CCN}$ was calculated by summing N elements above and to the right

of the constant $S_c$ curves. Contributions from the inorganic and organic particle types can also be calculated separately and then added to determine the total $N_{CCN}$ (i.e., $N_{CCN} = N_{CCN.inorg} + N_{CCN.org}$).





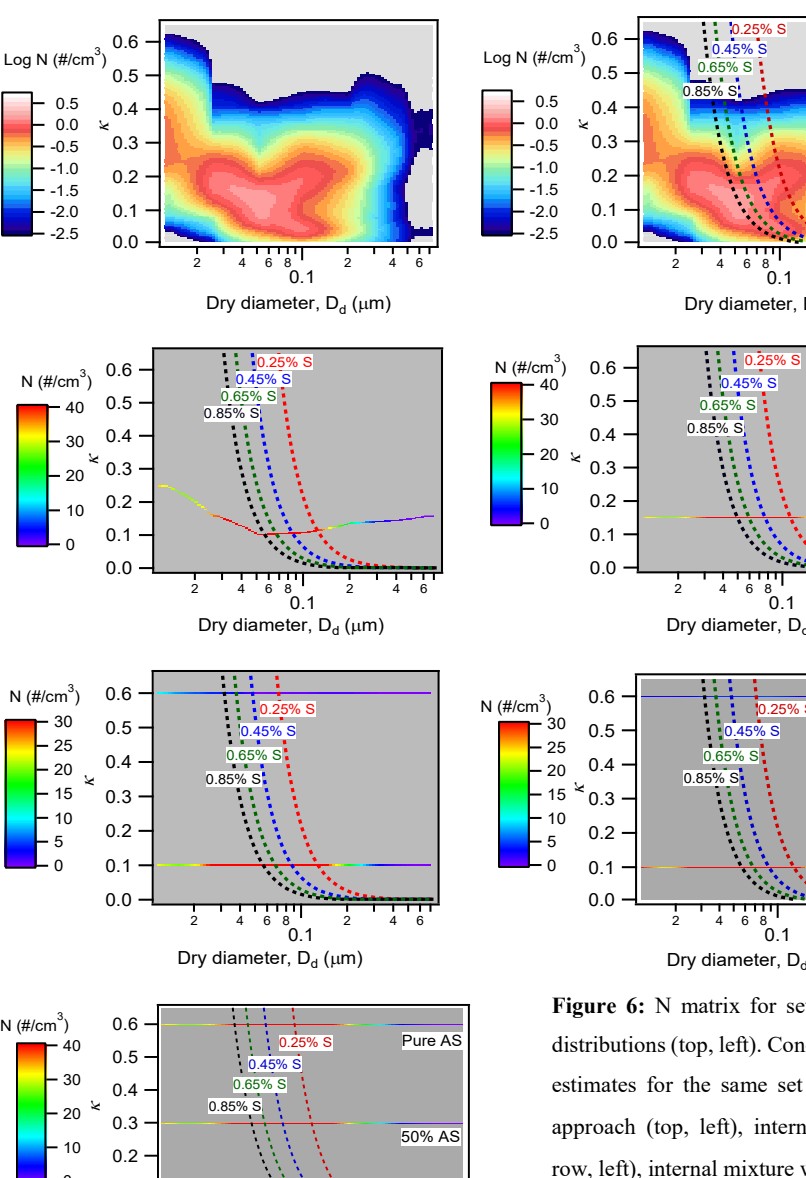

**Figure 6:** N matrix for set of measured size and GF distributions (top, left). Conceptual presentation of $N_{CCN}$ estimates for the same set of measurements: baseline approach (top, left), internal mixture with $\kappa(D_d)$ ($2^{nd}$ row, left), internal mixture with $\kappa(\neq D_d)$ ($2^{nd}$ row, right), external mixture with $\kappa(D_d)$ ($3^{rd}$ row, left), external mixture with $\kappa(\neq D_d)$ ($3^{rd}$ row, right), and internal mixture with assumed inorganic volume fractions (bottom).



# 6 Results and Discussion

### 6.1 Comparison between measured and baseline $N_{CCN}$ estimate

The concentration measured by the CCNc was compared to that calculated using the baseline approach for all available data from 2009 – 2012. The results for May, 2011 are shown in Figure 6 and for all of 2011 in Figure S3.

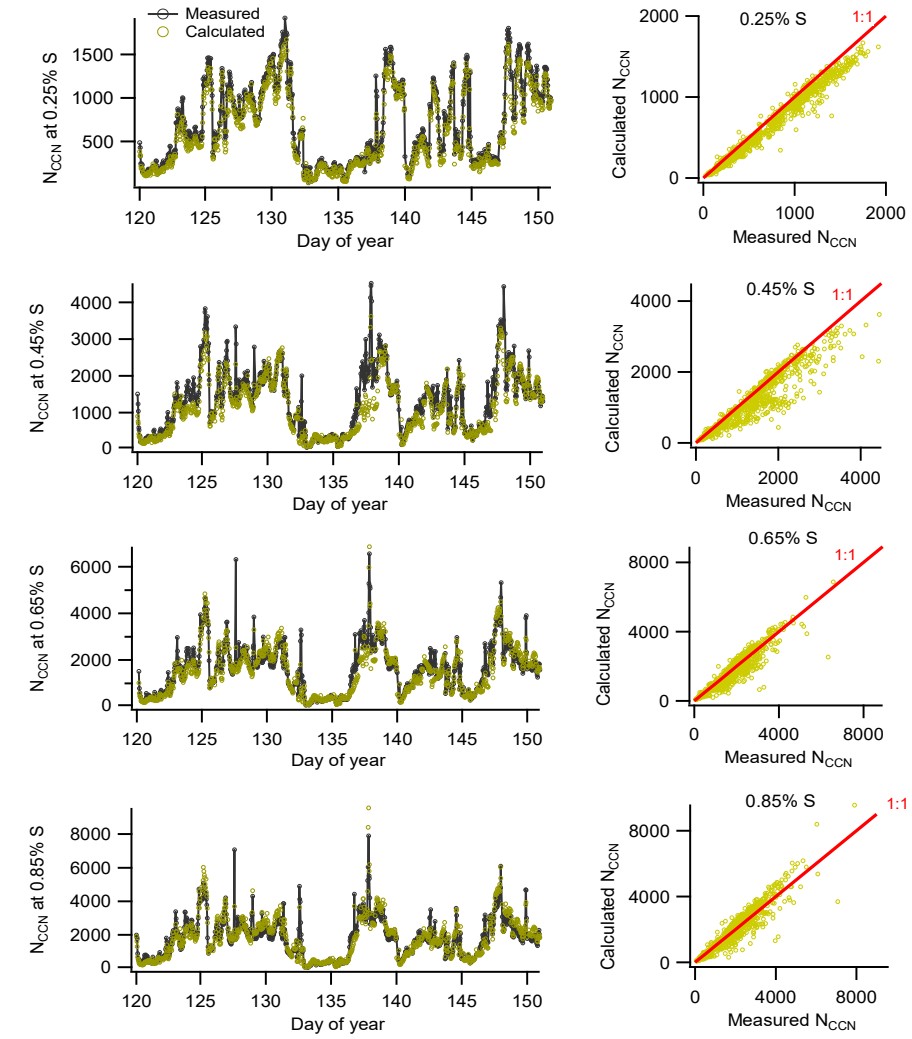

**Figure 7:** Comparison between measured and estimated $N_{CCN}$ (cm$^{-3}$) for May, 2011 at 0.25% (top row), 0.45% ($2^{nd}$ row), 0.65% ($3^{rd}$ row), and 0.85% S (bottom row).





The calculated $N_{CCN}$ tracks that measured throughout the period considered. The deviation between the two is quantified as the Normalized Root Mean Square Error (NRMSE)

$$NRMSE = \{\frac{1}{n}\sum_{i=0}^{n-1}\left(\frac{x_i - y_i}{x_i}\right)^2\}^{0.5} \qquad (6)$$

Where, $x_i$ is the measured $N_{CCN}$ at a given S, $y_i$ the estimated $N_{CCN}$ at the same S, and $n$ the number of concentration pairs

5    compared. The average NRMSE between the measured and calculated concentrations for May, 2011 and for each of the four years analyzed are summarized in Table 3. The measured $N_{CCN}$ at 0.25% S during 2012 was very noisy and was excluded from the analysis.

**Table 3.** NRMSE between measured and estimated $N_{CCN}$ at 4 different S from 2009 - 2012 and for May, 2011

| Year/ Month | NRMSE @ % S | | | |
|---|---|---|---|---|
| | 0.25 | 0.45 | 0.65 | 0.85 |
| 2009 | 0.53 | 0.36 | 0.35 | 0.35 |
| 2010 | 0.22 | 0.29 | 0.27 | 0.25 |
| **May, 2011** | **0.17** | **0.22** | **0.21** | **0.21** |
| 2011 | 0.29 | 0.29 | 0.26 | 0.26 |
| 2012 | - | 0.33 | 0.38 | 0.36 |

In addition to measurement error, some possible reasons for deviations between the measured and calculated concentrations are i) differences in assumed and actual properties of aerosol chemical species, ii) interactions among components not captured by $\kappa$-Köhler Theory, iii) the presence of low solubility organics that dissolve under the dilute conditions with S ~ $S_c$ but not in

15    the more concentrated solution in the HTDMA at 90% RH, iv) the presence of particles that contain slowly dissolving compounds or that are in an amorphous/glassy state for which hygroscopic growth and activation timescales may be comparable to or greater than the HTDMA and CCNc residence times, and v) the presence of surface tension-reducing species, which influence $S_c$ much more than GF.

**6.2 Comparison of $N_{CCN}$ calculated from different approaches**

20    Estimates of $N_{CCN}$ assuming the aerosol is an internal or external mixture relative to those of the baseline approach for which no assumption about mixing state is made are presented in Figures 8 and 9 for 0.25% and 0.85%, respectively, and in Figures S-4 and S-5 for 0.45% and 0.65% S, respectively. Best fits through the data were assumed to be linear and were forced through the origin to facilitate interpretation of the results and simply because of the apparent linear correlations with minimal offset in the figures. Table 4 summarizes the slope ($m = \frac{N_{CCN \text{ from alternate approach}}}{N_{CCN \text{ from baseline approach}}}$) and goodness of fit ($r^2$) for each of the different





approaches with respect to the baseline estimate. Values of $m$ above (below) 1 indicate that the mixing state assumption results in $N_{CCN}$ greater (less) than that from the baseline estimate for a given S.

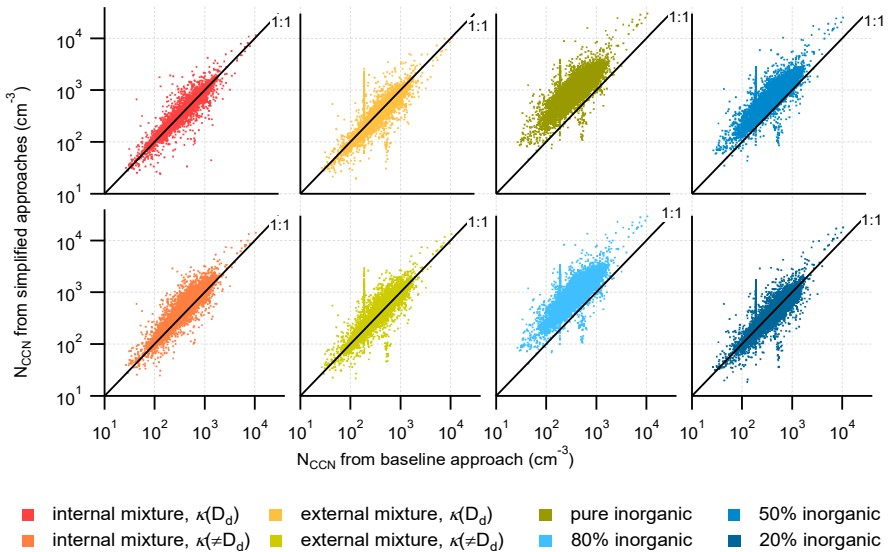

**Figure 8:** $N_{CCN}$ estimated from simplified approaches vs. baseline approach at 0.25% for all 2011 data.

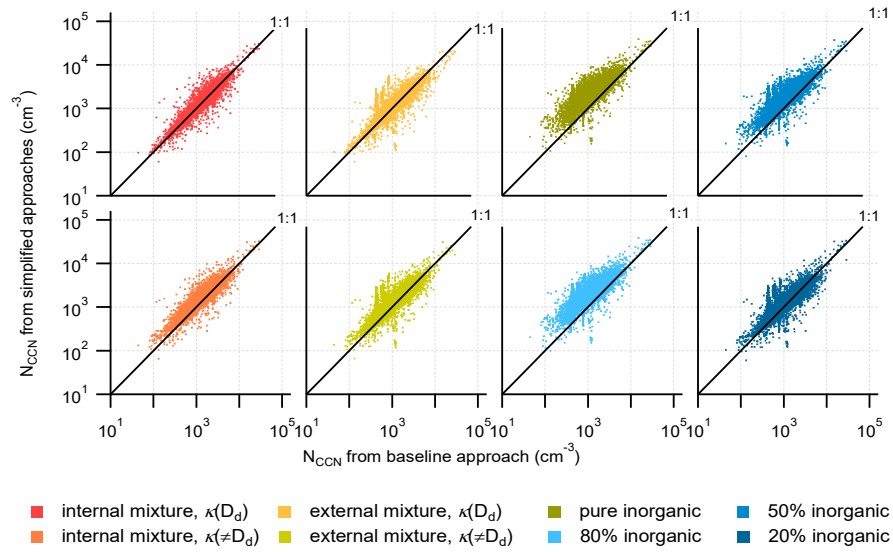

**Figure 9:** $N_{CCN}$ estimated from simplified approaches vs. baseline approach at 0.85% for all 2011 data.



**Table 4.** Fit parameters of $N_{CCN}$ estimate approaches for 2011 data

| Model | Slope ($m$) @ % S | | | | $\dfrac{m_{0.25\% S} - m_{0.85\%}}{m_{0.25\% S}}$ | Correlation coefficient ($r^2$) @ % S | | | |
|---|---|---|---|---|---|---|---|---|---|
| | *0.25* | *0.45* | *0.65* | *0.85* | | *0.25* | *0.45* | *0.65* | *0.85* |
| Baseline | 1 | 1 | 1 | 1 | 0 | 1 | 1 | 1 | 1 |
| Internal, $\kappa = \kappa(D_d)$ | 1.014 | 1.016 | 1.014 | 1.012 | 0.002 | 0.87 | 0.87 | 0.85 | 0.84 |
| Internal, $\kappa = \kappa(\neq D_d)$ | 1.214 | 1.181 | 1.132 | 1.099 | 0.095 | 0.82 | 0.82 | 0.81 | 0.8 |
| External, $\kappa = \kappa(D_d)$ | 0.912 | 0.927 | 0.909 | 0.9 | 0.009 | 0.72 | 0.74 | 0.73 | 0.73 |
| External, $\kappa = \kappa(\neq D_d)$ | 1.075 | 1.074 | 1.03 | 1.006 | 0.065 | 0.72 | 0.74 | 0.74 | 0.73 |
| Internal, pure AS | 2.39 | 1.83 | 1.572 | 1.435 | 0.4 | 0.65 | 0.66 | 0.66 | 0.65 |
| Internal, 50% AS | 1.77 | 1.5 | 1.34 | 1.262 | 0.284 | 0.69 | 0.73 | 0.73 | 0.72 |
| Internal, 20% AS | 1.06 | 1.06 | 1.034 | 0.999 | 0.058 | 0.7 | 0.75 | 0.75 | 0.74 |

The dependence of the best fit slopes on S is provided in the 6[th] column in Table 4 as a fractional change over the full range in

S considered. The utility of any of the simplifying approaches is obviously greatest if applicable over a wide range in S. The

fractional difference in m between 0.25% and 0.85% S is largest for the assumption of a pure AS aerosol and smallest when

the aerosol is assumed to be an internal mixture with $\kappa = \kappa(D_d)$. The assumption of an internal mixture with $\kappa = \kappa(D_d)$ also

resulted in the highest $r^2$. Based on the best fit slopes, the mean error introduced when using averaged $\kappa$ and assuming the

aerosol to be either an internal or external mixture varies from 1.4% to 16%. The balance between the importance of predictive

skill as reflected in these values and the computational efficiency will of course vary across applications.

**7 Summary**

Size distributions and hygroscopic growth factor distributions measured from 2009 to 2012 at the SGP ARM site were used to

estimate CCN concentrations over a range in supersaturation. An initial estimate of $N_{CCN}$ that served as a basis for comparison

used all of the information in the combined distributions without any averaging. For those estimates, matrices of $N(D_d, \kappa)$ and

$S_c(D_d, \kappa)$ were calculated from the measured distributions and from $\kappa$-Köhler Theory, respectively, and $N_{CCN}(S)$ was calculated

by integrating all of the N elements for which the corresponding $S_c$ element < S. Comparisons of those estimates with direct

measurements from a collocated CCN counter show that this baseline approach can reasonably predict $N_{CCN}$ over a range of

S.

The baseline spectra were then compared with those calculated using the same dataset but with aerosol treatments that are

more commonly used for more efficient computation or simply because size-dependent composition or hygroscopicity




distributions are not available. These included approximating the aerosol as an internal mixture with fixed inorganic volume fraction, as an internal mixture with size-dependent or size-independent hygroscopicity, and as an external mixture with size-dependent or size-independent hygroscopicity. Bias and variance relative to the baseline estimates were described with best fit slopes, $m$, and coefficients of determination, $r^2$, respectively, with both calculated from the thousands of $N_{CCN}$ pairs over the

period of analysis. Of the simplified treatments considered, assuming the aerosol is an internal mixture with size dependent hygroscopicity resulted in estimates closest to those from the baseline approach. These findings are strengthened by the use of a large dataset, but they are still most applicable for an aerosol similar to that found at SGP. Similar assessments are needed for other regions where aerosol characteristics such as composition and mixing state differ from those at SGP.

**8 Data availability**

All CCN concentration, chemical composition, size distribution, and hygroscopic growth factor distribution data used in our analysis were downloaded from the DOE ARM data archive at http://www.archive.arm.gov/. Derived products used for the comparisons provided in the manuscript can be obtained by contacting Dr. Don Collins (dcollins@tamu.edu).

**Acknowledgments**

This research was supported by the office of biological and environmental research of the U.S. Department of Energy under

grant DE-SC0016051 as part of the Atmospheric Radiation Measurement (ARM) climate research facility, an office of science scientific user facility. Data were obtained from the ARM climate research facility.

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
