# Peer review of "Influence of common assumptions regarding aerosol composition and mixing state on predicted CCN concentration"

_Atmospheric Chemistry and Physics, 2017_

## Referee Comment (RC1) · Anonymous Referee #1 · 20 Jul 2017

The manuscript "Influence of common assumptions regarding aerosol composition and mixing state on predicted CCN concentration" by Mahish et al. presents the analysis of a multi-year dataset collected at a mixed land use site in Oklahoma in an attempt to examine the accuracy of the assumptions commonly used by the modelling community to study and predict the number of cloud condensation nuclei NCCN concentration. The analysis employs the $\kappa$-Köhler theory and data collected by a variety of relevant instrumentation. At this point I, unfortunately, have to recommend this paper to be rejected from the publication by the Atmospheric Chemistry and Physics journal due to the reasons outlined below.

[Figure]

In its current state the manuscript is of little relevance and utility for the atmospheric and CCN community as it does not present anything new, and the conclusions it does present are not well-explained or justified. The effect of mixing state on aerosol CCN activity has been examined in numerous previous publications, and the authors present a multitude of relevant references on the second page of the manuscript. The authors do not provide any information on how their study is different from the published ones, do not clearly state their objectives taking into account the already existing knowledge, and, therefore, fail to convince me that the presented study is new or important. It has long been known that aerosol mixing state plays a minor role in determining the ambient CCN and, even more so, cloud droplet number concentration CDNC, especially so in non-pristine regions (Moore et al., 2013). The effects of the total particle number and the size distribution are of much higher importance than the particle hygroscopicity or the mixing state (e.g. Conant et al., 2004; Dusek et al., 2006). On page 2, lines 21‒24, the authors reference previous studies that have shown that NCCN is most sensitive to the particle size distribution and that assuming an internally-mixed aerosol is sufficient for an accurate NCCN prediction. The main conclusion of the study by Mahish et al. is mostly identical to this abovementioned statement, demonstrating the absence of any novel aspects in the study and deeming it a mere repetition of the work that has already been done before.

In general, the paper reads as a last-minute effort at writing something for a publication. The paper is short and incomplete. The storyline is confusing, a problem exacerbated by the excessive segmentation of the text into many short sections. The objectives, methodology, and the results are all described in a superficial manner, preventing the reader from understating exactly what was done, why it was done and what the outcomes are. The authors fail to conduct an in-depth analysis of the data, put their results in perspective and convince the reader of the importance of their findings. The analysis performed is rather basic, not of the quality and complexity level that would warrant its publication in a research journal. The manuscript also contains several stylistic, grammatical and other random mistakes and omits many important references.

[Figure]

Unfortunately, the manuscript is not of the standard for a publication in a scientific research journal, and I, therefore, recommend the manuscript to be rejected from the publication in the Atmospheric Chemistry and Physics journal.

REFERENCES

- Conant, W. C., Vanreken, T. M., Rissman, T. A., Varutbangkul, V., Jonsson, H. H., Nenes, A., Jimenez, J. L., Delia, A. E., Bahreini, R., Roberts, G. C., Flagan, R. C. and Seinfeld, J. H.: Aerosol-cloud drop concentration closure in warm cumulus, Journal of Geophysical Research: Atmospheres, 109(D13), doi:10.1029/2003jd004324, 2004.

- Dusek, U., Frank, G. P., Hildebrandt, L., Curtius, J., Schneider, J., Walter, S., Chand, D., Drewnick, F., Hings, S., Jung, D., Borrmann, S. and Andreae, M. O.: Size Matters More Than Chemistry for Cloud-Nucleating Ability of Aerosol Particles, Science, 312(5778), 1375–1378, doi:10.1126/science.1125261, 2006.

- Moore, R. H., Karydis, V. A., Capps, S. L., Lathem, T. L., and Nenes, A.: Droplet number uncertainties associated with CCN: an assessment using observations and a global model adjoint, Atmos. Chem. Phys., 13, 4235-4251, 2013.
* * *

---

## Author Comment (AC1) · 26 Jul 2017

Response to first anonymous reviewer

We appreciate the opportunity to respond to the reviewer's criticism of this manuscript. The review focuses on the important question of whether the manuscript adds enough to what is already known about the link between aerosol characteristics and CCN concentration to warrant publication. It does not offer specific recommendations for improvement. Thus, this response will be rather brief.

Reviewer's Comment:

[Figure]

In its current state the manuscript is of little relevance and utility for the atmospheric and CCN community as it does not present anything new, and the conclusions it does present are not well-explained or justified. The effect of mixing state on aerosol CCN activity has been examined in numerous previous publications, and the authors present a multitude of relevant references on the second page of the manuscript. The authors do not provide any information on how their study is different from the published ones, do not clearly state their objectives taking into account the already existing knowledge, and, therefore, fail to convince me that the presented study is new or important. It has long been known that aerosol mixing state plays a minor role in determining the ambient CCN and, even more so, cloud droplet number concentration CDNC, especially so in non-pristine regions (Moore et al., 2013). The effects of the total particle number and the size distribution are of much higher importance than the particle hygroscopicity or the mixing state (e.g. Conant et al., 2004; Dusek et al., 2006).

Answer:

We don't disagree with the reviewer's statement about the importance of the size distribution. Even so, the results summarized in Table 4 for the 3 scenarios for which particle hygroscopicity was assumed to be constant over time (20%, 50%, and 100% ammonium sulfate) show potentially large bias (4 – 81% for fixed composition vs. 1.4% for the best performing approach) and variability (average $r2$ of 0.70 for fixed composition vs. 0.86 for the best performing approach). Our goal was not to conclude that such differences were or weren't excessive, but rather to simply highlight the tradeoffs.

Reviewer's Comment:

On page 2, lines 21-24, the authors reference previous studies that have shown that NCCN is most sensitive to the particle size distribution and that assuming an internally-mixed aerosol is sufficient for an accurate NCCN prediction. The main conclusion of the study by Mahish et al. is mostly identical to this abovementioned statement, demonstrating the absence of any novel aspects in the study and deeming it a mere

repetition of the work that has already been done before.

Answer:

We believe that consistency of the findings presented in this manuscript with what has been presented in other publications should not preclude publication. Aerosol characteristics, sampling instrumentation, and analytical techniques vary widely among the publications that we cite and (undoubtedly) among others we did not. There are two noteworthy differences between the dataset we worked with and those used for the 11 publications on page 2, lines 21 – 24, the reviewer argues make this work duplicative.

1) The use of size-resolved growth factor distributions as the basis for the descriptions of mixing state and hygroscopicity. More so than for the datasets on which those other publications were based, we had a description of the actual size-resolved mixing state (at least a description of the mixing state that matters for this sort of analysis). The focus of the manuscript was a comparison of the CCN spectra determined when directly using the hygroscopicity distributions with those when the distributions were in some way averaged. And though a comparison with directly measured CCN spectra was included, it was only meant to show consistency among the measurements and was not the basis for conclusions about the most suitable description of the aerosol or about the error introduced as the full details contained in the measurements were simplified and averaged in different ways. So even if we arrived at the same conclusion as some of the referenced publications, we reached it following a rather different approach, which we feel makes this complementary of other analyses and not redundant.

2) The approximate mean and range of the duration of the datasets on which those publications are based are 22 days and 2 weeks - 1.5 months, respectively. Here we used an almost continuous 4-year dataset from a site at which there is considerable variability in aerosol properties and concentration over multiple timescales. In fact, the dataset used in this analysis is longer than those of the noted 11 publications combined. Of course we realize that more data doesn't necessarily mean better results,

but at least for SGP or sites like it the measures of bias and variability we report are more representative than if we had instead used just a month or so of data as with most similar studies. If given the opportunity to revise the manuscript we will more clearly articulate how our dataset and analyses differ from those used for other publications.
* * *

---

## Referee Comment (RC2) · Anonymous Referee #2 · 29 Aug 2017

This paper describes several years of growth factor, CCN data, and chemical composition data collected at the SGP site. CCN concentrations are calculated from aerosol size distributions and constrained assumptions about aerosol hygroscopicity. Sensitivity calculations to the assumptions mixing state are presented.

The presented work analyzes an impressive amount of data in the form of a traditional CCN closure study. The data and modeling results are interesting and relevant to the readers of ACP. Taking the analysis at face value, the results are what one would reasonably expect. However, important details are missing and the manuscripts requires significant clarification. In addition, the manuscript is poorly written.

[Figure]

I concur with the points raised by referee #1. As written, it is not clear how the manuscript is on par, or advances the field relative to the cited studies in the introduction.

The most significant weakness is the complete absence of details on the instrumentation, measurement uncertainties, data quality, complexity of the data set, and data processing used. The description of the methods is insufficient to understand what the authors did, and what the uncertainties are. This prevents a critical evaluation of the manuscript. In it's current form, the manuscript is not suitable for publication.

After several attempts, I cannot make sense of section 4. It is unclear what the authors mean by an ideal solution. The derivation of kappa from experimental data makes no claim about ideality or not. In practice, kappa serves as an activity coefficient. Perhaps the authors seek to express that they use the kappa measured at RH = 90% to predict CCN activity? If so, the assumption is that the activity coefficient at RH = 90% is the same as at the composition of droplet activation (which also is often at non-ideal compositions). There is abundant set of studies that show prove that this is a good assumption and an equally abundant set of studies that show that this may be a bad assumption. Another possibility is that the authors refer to ideality as ZSR mixing. In either case, I do not understand how the data in Figure 2 relate to ideality. A better explanation is needed here.

Equation 2 is presumably derived from the ZSR mixing rule. However, the derivation and origin of that equation is unclear. A kappa_org is derived through averaging over mixing state within the population at a single size, then averaged over all sizes, and the parsed through several unceratin quantities, including OA and BC mass fraction, mass absorption efficiency, OA and BC density, and an assumed kinorg. Given these assumptions, it would be important to define some uncertainty on the derived estimate. How relevant is the volume weighted average kappa to the 40-80 nm kappa that likely drives CCN closure? Figure 6 in Mahish and Collins shows an increase from ∼0.1 to ∼0.2 between 50 and 400 nm. What size does the volume weighted average kappa

most correspond to (i.e. what is the volume weighted mean diameter)?

Justification for splitting up the distribution shown in equations (4) and (5) is unclear. I can follow the algorithm mathematically. However, how to these three partial size distributions for inorg, org, and insoluble relate to the actual mixing state? Does this algorithm reproduce the measured kappa distributions at the different dry diameters? If so, this needs to be shown. If not, then the algorithm seems a semi-arbitrary decomposition into partial size distributions.

In general, the experimental description lacks important details. While data downloaded from an archive may be partially quality controlled, the required analysis for this type of work must go well beyond the standard Q/A procedures. Well described quality controlled data sets are a prerequisite to closure calculations.

(a) ACSM: The manuscript doesn't even provide a reference. to the ACSM. No lower size limit of the aerodynamic lens is provided. The potential role of species that is not measured by the ACM (black carbon, dust) in contributing to composition is ignored. Other experimental uncertainties in deriving mass fractions are not mentioned. (E.g. was derived sulfate compared to a PILS? What are the calibration factors applied to data? What is the statistical uncertainty in the fractions?) The ACSM is used in equation (2). Is there mass closure/volume closure between the ACSM and the SMPS volume? How relevant is ACSM data for CCN activation and growth factors at small sizes?

(b) GF: No GF data were presented, nor is even a brief description about the HTDMA included. Some technical details about flow rates, flow ratios, data inversion, humidity calibration, data inversion, and data reduction can be gleaned from two cited references (Collins, 2010, and Mahish and Collins, 2017). However, after reading Collins 2010, it lists data quality flags. How were these used in the processing? The reader should have sufficient information that they can download the data and repeat the calculations as presented in the paper. Currently this is not possible.

(c) CCN: It's surprising that no information is provided about the CCN data at all in the paper, given that the main focus is one of CCN closure. How was the supersaturation calibrated? How were the data quality controlled other than removing data points with Nccn>Ncn? (Why is the CCN broken in this case, and not the CPC?) How was the instrument operated (other than the brief info in Table 1)? Absolute temperature and pressure changes affect the instrument supersaturation, were those accounted for in the processing? Was stability of the temperature gradient monitored? Was the droplet size distribution used to monitor data quality? Was flow stability verified and calibrated? Where inlet losses accounted for?

(d) The origin of equation (2) is unclear. Giving the expression the benefit of the doubt, how was BC measured? An instrument to measure BC is not listed in Table 1. Which instrument in Yang et al. (2009) was used? Or is it cited for the mass absorption efficiency? Furthermore, what is the uncertainty in kappa_org using this approach? It should be evaluated using error propagation. Also, the seasonal cycle shown here should be distinguished from, and compared with, the seasonal cycle from their previous paper (Mahish and Collins, 2017), where the authors show the size resolved seasonal cycle in kappa over more years from the same dataset.

The closure approach to replicate absolute CCN concentrations from size distribution data. To do this well a myriad of non-trivial experimental issues must be addressed with care, which is not obvious from this paper. The DMT CCN instrument is prone to significant particle loss at D < 100 nm. For example, Figure 2 in Hodas et al. (http://www.atmos-chem-phys-discuss.net/acp-2016-236/) shows a gradual decline in activated fraction between 100 and 40 nm for ammonium sulfate, even though 100% are expected. The decline is due to losses in the inlet. Therefore, the absolute CCN number needs to be loss corrected before comparing to an integrated number from a size distribution. It needs to be verified that number closure between the SMPS and CPC is achieved. Furthermore, it needs to be verified that the SMPS sizing is stable and accurate, and preferably, the supersaturation calibration should done on the same

SMPS system than the one used for closure to avoid absolute biases. (A 5% sizing difference between the calibration SMPS and sample SMPS results in a significant bias in CCN closure). Getting all of this right is challenging for a short-term campaign and much more challenging in multi-year semi-autonomous sampling site. Discussion of these effects, uncertainty analysis, and long-term stability analysis are needed.

Discussion of the results in the context of past CCN closure attempts should be included, beyond the cursory mention in the introduction. There is a clear evolution of these studies over time, with more recent studies generally showing a higher success rate, in part due to the improved understanding of the experimental difficulties. How does the closure attempt here compare to previous results? What new insights are gained from this dataset?

Other comments

"In several studies, particles composed entirely of organic species have been reported to be largely ineffective in droplet formation (Abbatt et al., 2005; Prenni et al., 2007)"

I am not sure what ineffective in droplet formation means. Prenni et al. Report data for SOA having kappa = 0.1, which corresponds the kappa_org derived here and elsewhere. Are the the aerosol here ineffective in droplet formation?

"in Figure 6 and for all of 2011 in Figure S3"

change to Figure 7 and S3

Eq. 6, Table 3: what is the interpretation of NRSME values? Obviously smaller is better. What other utility does that metric provide in weighing the different approaches? Please specify.

Fig. 7 I suggest to add +/- 20% or similar error lines to the scatterplots.

[Figure]